# Potential Utility of Liquid Biopsy as a Diagnostic and Prognostic Tool for the Assessment of Solid Tumors: Implications in the Precision Oncology

**DOI:** 10.3390/jcm8030373

**Published:** 2019-03-18

**Authors:** Roshni Ann Mathai, Ryali Valli Sri Vidya, B. Shrikar Reddy, Levin Thomas, Karthik Udupa, Jill Kolesar, Mahadev Rao

**Affiliations:** 1Department of Pharmacy Practice, Manipal College of Pharmaceutical Sciences, Manipal Academy of Higher Education, Manipal, Karnataka 576104, India; roshniannmathai1@gmail.com (R.A.M.); vallisrividyar@gmail.com (R.V.S.V.); reddyshrikar@gmail.com (B.S.R.); levin.thomas@learner.manipal.edu (L.T.); 2Department of Medical Oncology, Kasturba Medical College, Manipal Academy of Higher Education, Manipal, Karnataka 576104, India; udupa.karthik@manipal.edu; 3Department of Pharmacy Practice & Science, 567 TODD Building, 789 South Limestone Street, Lexington, KY 40539-0596, USA; Jill.kolesar@uky.edu

**Keywords:** liquid biopsy, circulating tumor cells (CTCs), circulating tumor DNA (ctDNA), exosomes, cancer diagnosis, cancer prognosis

## Abstract

Liquid biopsy is a technique that utilizes circulating biomarkers in the body fluids of cancer patients to provide information regarding the genetic landscape of the cancer. It is emerging as an alternative and complementary diagnostic and prognostic tool to surgical biopsy in oncology. Liquid biopsy focuses on the detection and isolation of circulating tumor cells, circulating tumor DNA and exosomes, as a source of genomic and proteomic information in cancer patients. Liquid biopsy is expected to provide the necessary acceleratory force for the implementation of precision oncology in clinical settings by contributing an enhanced understanding of tumor heterogeneity and permitting the dynamic monitoring of treatment responses and genomic variations. However, widespread implementation of liquid biopsy based biomarker-driven therapy in the clinical practice is still in its infancy. Technological advancements have resolved many of the hurdles faced in the liquid biopsy methodologies but sufficient clinical and technical validation for specificity and sensitivity has not yet been attained for routine clinical implementation. This article provides a comprehensive review of the clinical utility of liquid biopsy and its effectiveness as an important diagnostic and prognostic tool in colorectal, breast, hepatocellular, gastric and lung carcinomas which were the five leading cancer related mortalities in 2018.

## 1. Introduction

Decades of astounding cancer research has defined cancer to be a disease that involves mutations in the cell genome [1]. Random mutations generated due to the development of genomic instability in cancer cells is considered to be a prominent factor that could orchestrate and expedite the acquisition of various hallmarks of cancer such as sustaining proliferative signaling, evading growth suppressors, resisting cell death, enabling replicative immortality, inducing angiogenesis, and activating invasion and metastasis [2]. The GLOBOCAN 2018 has estimated the global cancer incidence and mortality to be at 18.1 million and 9.6 million cancer deaths respectively in 2018. Cancer accounted for being the second leading cause of death globally. Lung cancer (18.4%), colorectal cancer (9.2%), gastric cancer (8.2%), hepatocellular cancer (8.2%) and breast cancer (6.6%) accounted for the top five leading causes of cancer-related deaths globally in 2018 [3].

The idea of precision oncology or precision medicine of cancer has emerged as a powerful clinical strategy in oncology with the objective of providing the most accurate and effective treatment to each cancer patient based on the genetic profile of cancer and the individual. Due to the genetic diversity and the rapidly changing dynamics of genomic profiles among cancer patients, better treatment efficacy could be attained if cancer therapeutics could shift from a concept of ‘one-size-fits-all’ approach to an individual level tailored treatment strategy [4]. Precision oncology is expected to emerge as an initiative to tackle various obstacles faced in cancer management, such as unexplained drug resistances, genomic heterogeneity of tumors and lack of appropriate methods for monitoring responses to therapies [5]. One of the prime challenges for the clinical implementation of precision oncology is to identify and detect molecular biomarkers that could predict the prognosis, sensitivity or resistance to a specific single agent or combination therapies, or specific therapy-associated adverse drug reactions [6]. In this scenario, liquid biopsy has been recently gaining widespread attention globally as an alternative/complementary to tissue biopsy in the era of “cancer theranostics” by being a minimally invasive prognostic and diagnostic tool that can assess the genetic landscape of various solid tumors.

The present study focuses on the clinical utility of liquid biopsy in the top five cancers that were the leading causes of cancer-related mortality in 2018 as per the GLOBOCAN 2018. The study was formulated after an exhaustive literature search from 165 articles collected through Google Search, PubMed and databases such as World Health Organisation (WHO) and www.fda.gov using the keywords ‘liquid biopsy’, ‘CTCs’, ‘ctDNA’, ‘exosomes’, ‘cancer diagnosis and cancer prognosis’ in combination with the Boolean operators “AND/OR.” Only the studies conducted in human subjects were considered for this report.

Liquid biopsy has emerged as a revolutionary technique that is providing new perspectives and dimensions to the field of medical oncology. It consists of the detection and isolation of circulating tumor cells (CTCs), circulating tumor DNA (ctDNA) and exosomes, as a source of genomic and proteomic information in patients with cancer [7]. Surgical biopsies (SB) are still considered to be the “gold standard” for diagnosis and treatment choice for diseases of genetic involvement such as cancer [8]. However, they are associated with inherent deficiencies such as:Limited accessibility of tumor tissue during tissue biopsy increases the chance of false-negative results [9].Lack of information regarding spatial and temporal heterogeneity of the tumor [10].Genetic landscape of the tumor might change due to the emergence of treatment-resistant sub clones which were in fewer numbers in the primary tumors [10].Problems due to logistic and technical complications such as tissue storage leads to false- positive results thereby affecting the treatment decision and patient care [11].Risk of adverse effects to the patient from the biopsy procedure [9].High total cost [12].

These disadvantages make tissue biopsy an unfeasible option for long term monitoring. Liquid biopsy, obtained with a routine blood draw, overcomes most of the limitations of tissues and can provide rapid detection of the tumor genetics including de novo and resistant mutations [13]. This technique involves the analysis of circulating tumor DNA, cell-free DNA, exosomes, RNA (mRNA and microRNA) and circulating tumor cells (CTCs) in the body fluids to determine the mutational status [14]. The US FDA approved the first liquid biopsy test on 1 June 2016 for analysis of *EGFR* mutations in Non-Small Cell Lung Carcinoma (NSCLC) patients using cobas *EGFR* Mutation Test v2. The test involves the detection of *EGFR* mutations in lung cancer patients whose tumors have the exon 18 (G719X) substitutions, exon 19 deletions, exon 20 insertions and substitutions (T790M, S768I) and exon 21 substitutions (L858R, L861Q) and is indicated to select patients who are candidates for *EGFR* inhibitor therapy [15,16]. Table 1 provides a brief description of the advantages and disadvantages of liquid biopsy.

## 2. Molecular Tumor Targets of Liquid Biopsy

As mentioned above, the analysis of the genetic aberrations could be performed by liquid biopsy using biomarkers such as ctDNA, CTCs and exosomes.

### 2.1. ctDNA

DNA is constantly released into the circulation as fragments by apoptosis and necrosis of both cancerous and non-cancerous cells in our body [26,27]. If the DNA is released irrespective of the cell of origin, it is typically referred to as cfDNA and when it is released specifically by the cancerous cells, it is referred to as ctDNA. Mutations, copy number variations (CNVs), methylation changes or single-nucleotide variations (SNVs) harbored by ctDNA could be analyzed with high sensitivity and specificity. ctDNA is considered a better option when compared to archival tissue DNA in clinical scenarios where new biopsies are difficult to obtain [17]. cfDNA levels of healthy subjects range from 0 to 100 ng/mL of blood, with an average of 30 ng/mL, whereas in cancer patients they range from 0 to 1000 ng/mL of blood, with an average of 180 ng/mL [28]. In cancer patients, ctDNA represents only a small proportion of total cfDNA (varies from less than 0.1% to over 10%). This proportion depends upon the tumor burden, cancer stage, cellular turnover and response to therapy [29]. The amount of ctDNA increases with tumor growth [30]. It is estimated that patients with a tumor load of 100 g in size (≈3 × 10^10^ neoplastic cells) release 3.3% of the tumor DNA into the circulation on a daily basis [31]. ctDNA can be isolated from plasma, serum, ascites, breast milk, lymphatic and peritoneal fluids, bone marrow aspirates, urine, prostatic fluid, peritoneal lavage, sputum, cerebrospinal fluid, gastric juice, and biliary and stool samples [32].

### 2.2. CTCs

CTCs are intact tumor cells shed from both primary tumor sites and metastatic sites into the circulatory system [33]. The number of CTCs present in the blood is as low as one CTC per 10^6^–10^7^ leukocytes per milliliter of blood, with even lower numbers in the early stages of cancer [34]. The detection and isolation of the CTCs have been achieved by the technological advancements that studied and utilized the antigenic expression of the CTCs and their physical differences with the leukocytes [35]. CTCs represent a highly dynamic cell population, characterized by a high heterogeneity at the genetic, transcriptomic, proteomic and metabolomic levels [36]. The phenotypic and genotypic characteristics of CTCs can change during the course of the cancer by microenvironmental and therapeutic selective pressures [37]. As CTCs counts run in parallel with the tumor burden of the disease, they serve to be a more accurate method for the real time monitoring of cancers than many other commonly used soluble biomarkers [38,39,40].

### 2.3. Exosomes

Exosomes consist of a lipid bilayer which contains both transmembrane and nonmembrane proteins, as well as noncoding RNAs, mRNAs, and either single-stranded or double-stranded DNA [41]. Presence of exosomes was first reported in 1983 by Pan and Johnstone when culturing sheep reticulocytes at McGill University [42]. Exosomes are small membrane-enclosed vesicles 50–150 nm in size and 30–120 nm in diameter [43] which are actively discharged by most cells, including tumor cells [44] to extracellular space or biological fluids including serum [45], urine [46], breast milk, plasma, saliva [47], tears [48], pleural effusion [49], semen [50], amniotic fluid [51] and synovial fluid [52]. Studies have found that the analysis of double-stranded DNA [41] and RNA [53] content of exosomes provides details about the mutational status of the original cells as the architecture of exosomes protect RNA and miRNA from RNase catalytic activity thus providing accurate details of the primary tumor traits. A study has found that the integrin composition of exosomes promotes organotrophic metastasis [54] and thus the analysis of nucleic acids in exosomes can provide details regarding metastasis, tissue invasion and angiogenesis [55]. Isolation of exosomes from body fluids is carried out by technologies that are based on the biophysical properties (size, morphology, density), immunoaffinity capture or by precipitation method [56]. The clinical utility of exosomes as a tumor biomarker in cancer require further evidence from large clinical trials as most of the existing data are limited to small cohort studies [7].

## 3. Methods

Several methods have been proposed for the isolation and analysis of cfDNA, CTC and exosomes. cfDNA can be isolated using cfDNA isolation kits such as DNA isolation kit for mammalian blood (Boehringer Mannheim, now Roche Molecular Biochemicals, Mannheim, Germany), the QiaAmp blood kit (Qiagen, Germantown, MD, USA), and the Jetquick Blood Kit (Thermofischer, Waltham, MA, USA) which are available in the market [57]. Other methods used for cfDNA extraction include isolation with organic solvents such as phenol or chloroform [58] and the use of magnetic beads [59]. Isolation of cfDNA is followed by massive parallel sequencing [60,61] and digital genomic methods [62,63] for detection, quantification and molecular characterization of ctDNA fraction. CELLSEARCH^®^ is an FDA approved test for capturing and enumerating CTCs. This method uses ferrofluid reagent, which consists of particles with a magnetic core surrounded by a polymeric layer coated with antibodies targeting the Epithelial cell adhesion molecule (EpCAM) antigen, for capturing CTCs [64]. The CTC chip, a microfluidic device is one of the techniques used for isolation of CTCs. This chip contains an array of microposts that contain anti-epithelial-cell-adhesion-molecule (EpCAM, also known as tumor-associated calcium signal transducer 1 (TACSTD1)) antibodies. These antibodies have an affinity towards the EpCAM, which is over-expressed by the carcinomas of lung, colorectal, breast, prostate, head and neck and hepatic origin [65] thereby making it a useful tool in the CTC isolation process with high specificity [66,67]. Apart from microchips, filtrating systems such as VyCAP or ISET (Isolation by SizE of Tumor cells) filtration [68] which can be enhanced by the bead-based capture [68] are also used for CTC isolation. Exosome isolation by conventional methods such as Western blotting and enzyme linked immunosorbent assay (ELISA) require large sample volume and extensive processing [69,70]. Hence methods such as nano-plasmonic sensor [71], BEAMing and Droplet Digital PCR Analysis (ddPCR) [72], microfluidic exosome analysis [73], microfluidic devices such as ExoChip [74], SOMAmer^®^-based assay technology [75] and surface plasmon resonance imaging [76] were developed to overcome these challenges.

Isolation is followed by PCR amplification and next generation sequencing (NGS). Hypermethylation, hypomethylation, deletions, amplifications, chromosomal rearrangement and mutations can be detected by PCR and NGS technologies [77] as depicted in Figure 1. Scorpion amplified refractory mutation system (ARMS), allele-specific quantitative PCR, PCR with peptide nucleic acid clamps, massively parallel sequencing, and ddPCR [78], COBAS *EGFR* test, competitive allele-specific PCR [79] and mutant specific PCR [80] are the widely used PCR techniques. NGS allows the detection of multiple mutations in multiple genes in the following four steps: generation of short fragment DNA library, single fragment clonal amplification, massive parallel sequencing and data analysis [77]. In ultra-deep sequencing, the focus can be narrowed down on the clinically relevant gene panels and the mutations can be detected with high sensitivity and specificity [81,82]. AURA II studies have shown that the testing performance of NGS in detecting the *EGFR* mutation is relatively comparable to the PCR based and COBAS technologies [83]. NGS based *EGFR* mutation testing has shown to facilitate the determination of prognosis in patients with advanced NSCLC [84], monitoring genomic alterations [85,86,87], determining the resistance landscape to targeted therapies [88] and in predicting the response to therapy [89].

## 4. Significance of Liquid Biopsy in Various Cancers

Several studies have demonstrated that liquid biopsy could be used as a potential tool for the detection of genetic alterations in a wide variety of cancers. Studies have identified the presence of cancer-specific biomarkers such as *EGFR*, *KRAS*, *ERBB2, EML4-ALK, CEA, SEPT9* in the body fluids of patients and have shown the possibility of reconstructing the tumor genomes from plasma DNA [90,91,92,93,94]. The clinical utility of other biomarkers in cancer is mentioned in Table 2. As the primary objective of the therapy is to prevent tumor progression, metastasis and recurrence, proper monitoring is required to check if the therapy fulfills the above requirement. Since liquid biopsy is feasible to repeat during follow-up, it can be used for monitoring therapeutic response and prognosis [95]. Here, we describe the clinical utility of liquid biopsy in several cancers.

### 4.1. Colorectal Cancer

Colorectal cancer (CRC) is the third most common cancer worldwide and the second most prevalent cause of cancer deaths according to GLOBOCAN 2018 [96]. Mutations of genes such as *KRAS* [97], *BRAF* [98], *TP53* [99], *APC* [100], *CEA* [101] and *SEPT9* [102] are frequent in CRC. Detection of mutations in these genes by liquid biopsy is being studied as a cancer screening tool in populations at risk of cancer. Research studies have demonstrated that *APC* [29], *BRAF* and *KRAS* mutations were identified in ctDNA with high sensitivity and specificity [103]. When compared to breast cancer or prostate cancer, colon cancer patients had very low levels of CTCs in external circulation making their detection difficult [31]. Liquid biopsy has been investigated as a method to analyse colon cancer staging as well as prognosis. TNM staging is significantly correlated with tumor traits in the blood samples of the patients [104]. Further, the depth of tumor invasion also showed a significant correlation with the presence of biomarkers [105]. Additional evidence validating the detection of ctDNA, CTCs, and cfDNA as a marker for early diagnosis in CRC should be obtained. Usually, ctDNA and CTC [105,106] levels are found to be associated with poor prognosis.

cfDNA concentration was increased in CRC patients and elevated cfDNA levels were also associated with poor prognosis [107]. Patients with three or more CTCs/7.5 mL are found to have a reduced survival [108]. Yet, a study by Bessa et al. found no correlation between CTC levels and prognosis in postoperative CRC patients [109]. An increase in cfDNA levels has been observed in patients who had a recurrence of mutation when compared to those with loss of mutation [110].

### 4.2. Breast Cancer

Breast cancer (BCa) is the second most common cancer worldwide and the fifth most prevalent cause of cancer deaths according to GLOBOCAN 2018 [96]. ctDNA of BCa patients contained the somatic SNVs, CNAs [90] thereby making it a useful tool for monitoring tumor burden [118], screening, understanding the drug response, determining prognosis [119,120,121] and detecting minimal residual disease [122]. Tumor size, lymph node metastasis, stage and grade were found to have a close relation with ctDNA among BCa patients [123,124,125,126,127]. cfDNA levels were found to be low in patients with non-metastatic breast disease when compared to malignant breast disease [128]. Studies have suggested that HER2 receptor status can be assessed using liquid biopsy technique. A retrospective study done on 107 CTC positive metastatic BCa patients depicted that liquid biopsy could be a useful method for revaluation of HER2 receptor status [111]. A case report by Tzeng et al. described that liquid biopsy was superior to IHC in determining the HER2 status [129]. Mayor et al. had reported that *BRCA1* gene mutation could also be identified using liquid biopsy [112]. Several studies have shown the capacity of ctDNA in identifying the tumor-derived genomic alterations in BCa patients [130]. Dawson et al. demonstrated the presence of ctDNA and CTC in 97% and 87% of malignant BCa patients respectively [121]. A study reported a reduction in cfDNA integrity in metastatic BCa patients when compared to primary BCa patients [131]. Aceto et al. demonstrated that CTC clusters have up to 50 times increased metastatic potential when compared to individual CTCs [132]. Recurrence is a major problem in BCa. Liquid biopsy can be used as a potential tool for detecting tumor recurrence as tumor-specific copy number aberrations persist to about 12 years after diagnosis [129]. Liquid biopsy has been projected to be a potential diagnostic tool for determining the resistance to therapy [133].

### 4.3. Hepatocellular Carcinoma

Hepatocellular carcinoma (HCC) is the sixth most common cancer worldwide and the fourth most prevalent cause of cancer related deaths according to GLOBOCAN 2018 [96]. In all HCCs and specifically in small tumors located near the diaphragm, liquid biopsy is considered to be a preferable option as the tumors are not easily accessed by fine needle biopsy. Many findings have reported that the occurrence of genetic mutations in plasma, serum and urine samples of patients with HCC [113]. Hepatocytes, cholangiocytes and hepatic stellate cells can act as both exosome releasing or targeting cells [134]. A study on 14 advanced HCC patients has shown that ctDNA can be utilized as a diagnostic marker in the detection of *TP53, CTNNB1, PTEN, CDKN2A, ARID1A, MET, CDK6, EGFR, MYC**, BRAF, RAF1, FGFR1, CCNE1, PIK3CA* and *ERBB2/HER2* mutations [113,114,115]. The role of exomes in the detection of the mutations in HCC is well established. The exosome content derived from HCC and non-tumor liver cells varied significantly. Exosomal mRNAs such as miR-21, miR-18a, miR-221, miR-222 and miR-224 serve as biomarkers in HCC [135,136,137,138,139]. Apart from exomes, CTCs are also found to be a suitable alternative. Tumor invasion, tumor size, differentiation status, the disease extent and survival [136], were significantly associated with CTC levels [140]. A major relation between the number of circulating cancer stem cells and intrahepatic and extrahepatic recurrence was observed, thus suggesting its role as a sovereign marker of survival [141]. Several studies have found that liquid biopsy was also found to be helpful in the early detection of HCC. DNA copy number aberrations were found in two HBV carriers without previous history of HCC during blood collection. Upon reassessment, two patients developed HCC, thereby shedding light on the evaluation of copy number aberrations in ctDNA as a screening tool for early HCC detection [142].

### 4.4. Gastric Cancer

Gastric cancer (GC) is the fifth most common cancer worldwide and the third most prevalent cause of cancer deaths according to GLOBOCAN 2018 [96]. Ling et al. reported that methylated XAF1 DNA was found in 69.8% (141/202) of GC patients and none in healthy individuals, thereby serving as a potential diagnostic and prognostic marker [143]. A study by Park et al. [144] reported a high plasma MYC/GAPDH ratio in GC patients when compared to healthy individuals. Kang et al. [145] demonstrated the significance of plasma hTERT mRNA as a potential diagnostic and prognostic marker in GC. CTCs are found to be effective in the detection of cancer-specific modifications such as altered expression of non-coding RNAs (e.g., miRNAs) in GC [146]. Studies have reported a lower overall survival rate in metastatic GC patients with higher CTC levels [147,148]. Results of a study that was conducted by Shoda et al. to detect *HER2* amplification in cfDNA using RQ-PCR showed that cfDNA could be used as a significant therapeutic biomarker in the diagnosis and assessment of *HER2* status [149]. Shoda et al. demonstrated a correlation between plasma and tissue *HER2* amplification ratios by ddPCR [116]. Wu et al. [117] have shown that simultaneous sensitivity assay of the combination of markers such as *TERT, CK19, CEA* and *MUC1* using a high-throughput colorimetric membrane array provides a platform for assessing the overall survival and postoperative recurrence/metastasis. Mimori et al. [150] demonstrated that membrane MT1-MMP mRNA levels in the peripheral blood serves as a prognostic indicator for determining recurrence and distant metastasis. Programmed death-ligand 1 (PD-L1) mRNA expression in the blood of advanced GC patients is significantly higher than that of early GC patients, suggesting its utility in assessing the prognosis [151]. The PD-L1 expression significantly correlated with the depth of invasion, metastasis and stage of cancer.

### 4.5. Lung Cancer

Lung cancer (LC) is the most common cancer worldwide and the most prevalent cause of cancer deaths according to GLOBOCAN 2018 [96]. Several genomic alterations have been identified in advanced LC patients using liquid biopsy that could aid in the determination of prognosis [152]. Couraud et al. have shown the utility of liquid biopsy based on detection of *EGFR, KRAS*, *BRAF*, *ERBB2*, *PIK3CA* mutations using cfDNA [81,153]. The study by Paweletz et al. demonstrated the detection of *ALK*, *ROS1*, and *RET* rearrangements, *HER2* insertions, and *MET* amplification in patients with advanced NSCLC [77]. Newman et al. demonstrated that cfDNA levels in LC patients significantly correlated with CT and PET measured tumor volume. This study also revealed a correlation between ctDNA levels and tumor volume [30]. Liquid biopsy also serves as a guiding tool for estimating the response of targeted therapy [154]. ctDNA levels can be utilized in tracking the subclonal nature of NSCLC relapse and metastasis [155]. Sozzi et al. showed that higher plasma DNA levels are associated with reduced 5-year survival [156]. The CTC count was found to be helpful in the determination of prognosis and survival time [157]. Multiple studies have demonstrated that patients with high levels of CTCs at initial diagnosis or after one cycle of chemotherapy showed a poor LC prognosis [158]. CTCs could serve as a surrogate marker of distant metastasis in patients with primary LC [159]. Meta-analysis of 12 randomized control trials has shown that the presence of *KRAS* mutation correlated well with the lower survival rate in NSCLC patients [160]. However, another study conducted by Camps et al. on patients with complex NSCLC demonstrated no relationship between *KRAS* mutation and prognosis [161]. Recurrence of LC has been observed in patients who had detectable levels of CTCs in their blood [162]. cfDNA is now proposed to be a useful tool in determining resistance to therapy [163].

## 5. Conclusions

The utilization of CTCs, ctDNA and exosomes as potential biomarkers for cancer theranostics is an emerging area with a strong potential for clinical utility. Liquid biopsy is emerging as a minimally invasive, repeatable and inexpensive method for accessing the tumor DNA, understanding the tumor heterogeneity, monitoring therapeutic effectiveness, prognosis, acquired resistance to therapy and disease resistance in cancer. Further large scale studies should be conducted to validate the process and assess its clinical utility in different populations. Currently, liquid biopsies have limited applications in clinical practice, but its versatility and advantages put forward its application as a promising diagnostic and prognostic tool for precision oncology.

## Figures and Tables

**Figure 1 jcm-08-00373-f001:**
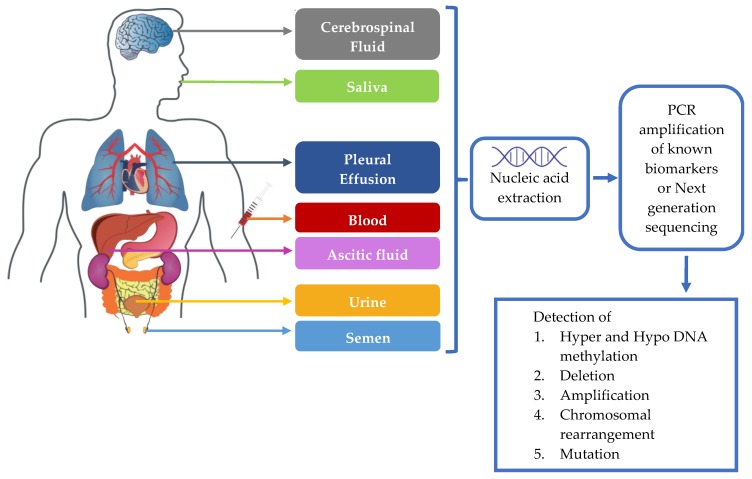
Procedure for performing liquid biopsy.

**Table 1 jcm-08-00373-t001:** Advantages and disadvantages of liquid biopsy.

Advantages	Disadvantages
Helps in understanding the spatial and temporal heterogeneity of cancer [17]. Requires only a small amount of blood (usually 6–10 mL of blood) [18]. Minimally invasive [19]. Early detection of cancer [20]. Real time monitoring for treatment responses and resistance could be performed by repeated analysis [21]. Shorter turnaround time for genotyping mutations [18]. Aid precision oncology [22].	Lack of standardization of the techniques [23]. Sufficient clinical and technical validation is not yet attained, that is required for the routine clinical implementation [24]. In some cancers (e.g., lung cancers), the diagnosis and subtyping cannot be done by liquid biopsy and can be established by only histology [25].

**Table 2 jcm-08-00373-t002:** Utility of liquid biopsy in various cancers.

Sl.no	Cancer Type	Genes	Reference
1.	Colorectal cancer	*KRAS, BRAF, TP53, APC, CEA, SEPT9*	[97,98,99,100,101,102]
2.	Breast cancer	*HER2, BRCA1*	[111,112]
3.	Lung cancer	*KRAS, EGFR, BRAF, ERBB2, PIK3CA, ALK, ROS1, RET, HER2, MET*	[77,81]
4.	Hepatocellular cancer	*TP53, CTNNB1, PTEN, CDKN2A, ARID1A, MET, CDK6, EGFR, MYC, BRAF, RAF1, FGFR1, CCNE1, PIK3CA, ERBB2/HER2*	[113,114,115]
5.	Gastric cancer	*MUC1, CK19, HER2* *TERT, CEA*	[116,117]

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
