# Peer review of "Potential Utility of Liquid Biopsy as a Diagnostic and Prognostic Tool for the Assessment of Solid Tumors: Implications in the Precision Oncology"

_jcm, 2019, doi:10.3390/jcm8030373_

Round 1
Reviewer 1 Report
The manuscript by Roshni Ann Mathai and colleagues with the title “Potential Utility of Liquid Biopsy as a Diagnostic and Prognostic Tool for the Assessment of Solid Tumors: Implications in the Precision Oncology” is an attempt to grasp to huge field of liquid biopsies. Although it is of great importance to summarize novel findings in liquid biopsies, the manuscripts lacks several important points. I addressed some of the missing point below:
In general, several references are just websites. These may change in short time. Always search for the peer reviewed papers.
In general, the language needs extensive editing
In general, the selection of tumor entities to be included in the review for a detailed discussion is very random. Was it based on the top 10 tumor incidences?
Line 30: The definition of cancer is not very comprehensive and there is no reference cited. Further, several hallmarks are missing, please read and cite a proper paper, such as Hallmarks of Cancer: The Next Generation, Hanahan, Weinberg, Cell 2011
Line 33: The definition of personalized medicine is not correct. It is broader than the individual somatic mutation profile.
Line 45: There is general bias for the used search strategies for this review. For example, www.cellsearchctc.com is a company-owned and should not be used as a search platform. It is just a technology. If using a website, always go back to the original peer reviewed literature.
Line48: Defnition of liquid biopsy is insufficient (eg only genetic landscape described). It is much more, and the reference is missing.
Table 1 and line 36: The term “precision therapies” is often used. There is no red threat for the terms precision and personalize.
Table 1: The disadvantages are unclear, eg: point 4 the predictive performance of …, point 5 is very specific for a tumor entitiy.
Line 78 – Line 81 please add a valid reference. Otherwise, not valid.
Line 85: Suddenly the CTC are described as source o cf/ctDNA. This part can be very puzzling for the reader as before the authors stated that the source of cf/ctDNA is the tissue.
Line 88: The SNV weblink is still in the manuscript, but not cited!
Line 96: CTC as source for ctDNA levels, please elaborate on that
Line 125-131 Methods are very little described. Either refere to another good revie describing methods, or state the most important ones.
Line 133: PCR and NGS are not just used for methylation status. also for SNvs etc… elaaorate on that point
Line 153:Which utility? Clinical utility?
Line 188: What are SVs? If using abbreviations for the first time, you need to state the full term first.
Author Response
Response to Reviewer 1 Comments
The manuscript by Roshni Ann Mathai and colleagues with the title “Potential Utility of Liquid Biopsy as a Diagnostic and Prognostic Tool for the Assessment of Solid Tumors: Implications in the Precision Oncology” is an attempt to grasp to huge field of liquid biopsies. Although it is of great importance to summarize novel findings in liquid biopsies, the manuscripts lacks several important points. I addressed some of the missing point below:
In general, several references are just websites. These may change in short time. Always search for the peer reviewed papers.
Response: We have deleted most of the references that are websites in the modified manuscript. We have only one website reference i.e., FDA. This FDA website was used for extracting the literature pertaining to the first FDA approved liquid biopsy test literature.
In general, the language needs extensive editing
Response: The comments were duly noted and extensive language editing was done across the manuscript.
In general, the selection of tumor entities to be included in the review for a detailed discussion is very random. Was it based on the top 10 tumor incidences?
Response: Thank you for the valuable suggestion given to us. We have modified our manuscript to include the significance of liquid biopsy in colorectal cancer, breast cancer, hepatocellular cancer, Gastric cancer and lung cancer that were the top 5 five leading cancer related mortalities as per GLOBOCAN 2018. Hence, during our manuscript editing we have excluded the significance of liquid biopsy in pancreatic cancer and in other cancers.
Line 30: The definition of cancer is not very comprehensive and there is no reference cited. Further, several hallmarks are missing, please read and cite a proper paper, such as Hallmarks of Cancer: The Next Generation, Hanahan, Weinberg, Cell 2011
Response: As per your invaluable suggestion, we have included the highly cited two literatures of Hanahan and Weinberg- The Hallmarks of Cancer and Hallmarks of Cancer: The Next Generation as our 1st and 2nd references in our introduction section.
Line 33: The definition of personalized medicine is not correct. It is broader than the individual somatic mutation profile.
Response: As per your suggestion, the term ‘personalized medicine’ has been replaced by ‘Precision medicine’ or ‘Precision oncology’ across the entire manuscript.
Line 45: There is general bias for the used search strategies for this review. For example, www.cellsearchctc.com is a company-owned and should not be used as a search platform. It is just a technology. If using a website, always go back to the original peer reviewed literature.
Response: We have removed the website reference www.cellsearch.com and replaced with a peer reviewed literature.
Line48: Defnition of liquid biopsy is insufficient (eg only genetic landscape described). It is much more, and the reference is missing.
Response: We have redefined liquid biopsy with an appropriate reference.
Table 1 and line 36: The term “precision therapies” is often used. There is no red threat for the terms precision and personalize.
Response: As per your suggestion, the term ‘personalized medicine’ has been replaced by ‘Precision medicine’/ ‘Precision oncology across the entire manuscript.
Table 1: The disadvantages are unclear, eg: point 4 the predictive performance of …, point 5 is very specific for a tumor entitiy.
Response: As per your and reviewer 2 suggestions, we have removed 4th and 5th disadvantages and the other disadvantages have been rephrased citing with appropriate references.
Line 78 – Line 81 please add a valid reference. Otherwise, not valid.
Response: We have removed the text owing to your comments.
Line 85: Suddenly the CTC are described as source o cf/ctDNA. This part can be very puzzling for the reader as before the authors stated that the source of cf/ctDNA is the tissue.
Response: As per your suggestion, CTC as the source of cf/ctDNA, can be very puzzling for the reader. Hence, we have rephrased the clinical utility of CTC on the basis of CTC count as a molecular tumour target.
Line 88: The SNV weblink is still in the manuscript, but not cited!
Response: We have removed SNV weblink from the document.
Line 96: CTC as source for ctDNA levels, please elaborate on that
Response: As per your suggestion, CTC as the source of cf/ctDNA, can be very puzzling for the reader, we have rephrased the clinical utility of CTC on the basis of CTC count as a molecular tumour target.
Line 125-131 Methods are very little described. Either refere to another good revie describing methods, or state the most important ones.
Response: We have now described and rewritten the most important liquid biopsy methodologies.
Line 133: PCR and NGS are not just used for methylation status. also for SNvs etc… elaaorate on that point
Response: In addition to methylation status, we have rephrased the line to include deletions, amplifications, chromosomal rearrangement, and mutations that could be detected by PCR and NGS technologies.
Line 153: Which utility? Clinical utility?
Response: In the line no 153, the term “utility is replaced with ”clinical utility”.
Line 188: What are SVs? If using abbreviations for the first time, you need to state the full term first.
Response: In the current modified manuscript the term “SVs” is excluded.
Reviewer 2 Report
Major:
Liquid biopsy techniques have tremendous potential in the cancer diagnosis, therapy and assessments but still in development. Sensitivity and specificity are major concerns, and validations are needed. Those points are covered reasonable well in the body and the conclusion of the article. However, the abstract did not point those facts out. Therefore, the abstract should be rewritten to emphasize ‘the potential future’ and balance ‘current limitation’ as well.
Minors:
1) Page 1, line 33: ‘Personalized medicine is a NOVEL approach…’ Suggest changing ‘novel’ to ‘idea’.
2) Page 1, line 37-38: ‘While assessment of the somatic mutation profile is considered a standard of care for many cancers,…’ –based on limited actable mutations in most cancers, assessment of the somatic mutation profile has NOT be standard care yet..
3) Page 2, Table 1, under ‘Disadvantages’: ‘Changes of cancer cells to small cell cancer cannot be identified using current liquid biopsy technique’ – Does not make sense. What dose ‘small cell caner’ mean?
4) Page 3, line 83: ‘Several studies have…” Need newer references for the statement (the current references were too old, from 1975, 1977).
5) Page 3, line 90-91: ‘A study conducted by Bettegowda C et al has demonstrated that ctDNA could be detected in patients without detectable CTCs’. Suggest changing to ‘A study conducted by Bettegowda C et al has showed that ctDNA could be detected in patients without images detectable CTCs’.
6) Page 3, line 98-100: “There is currently… inform prognosis” Need a reference.
7) Page 5, line 175 : ‘benign’ should be ‘non-metastatic’.
8) Page 5, line 178-181: The sentence ‘A study on… disease[97]’ is confusion, needs to redo.
9) Page 6, line 210-212: ‘Recurrence…As …Detection’ needs to redo
Author Response
Response to Reviewer 2 Comments
Major:
Liquid biopsy techniques have tremendous potential in the cancer diagnosis, therapy and assessments but still in development. Sensitivity and specificity are major concerns, and validations are needed. Those points are covered reasonable well in the body and the conclusion of the article. However, the abstract did not point those facts out. Therefore, the abstract should be rewritten to emphasize ‘the potential future’ and balance ‘current limitation’ as well.
Response: Thank you for your invaluable suggestion, we have rewritten the abstract to emphasize the potential future and current limitations of liquid biopsy. As raised by you, sensitivity and specificity and the need for validations requiring large scale studies were incorporated to the modified abstract.
Minors:
1) Page 1, line 33: ‘Personalized medicine is a NOVEL approach…’ Suggest changing ‘novel’ to ‘idea’
Response: As per the suggestions of the reviewer 1, the term “personalized medicine” has been replaced with “precision medicine/ precision oncology” across the manuscript.
2) Page 1, line 37-38: ‘While assessment of the somatic mutation profile is considered a standard of care for many cancers,…’ –based on limited actable mutations in most cancers, assessment of the somatic mutation profile has NOT be standard care yet..
Response: As per your valuable suggestion, we have removed the lines 37-38 from the manuscript.
3) Page 2, Table 1, under ‘Disadvantages’: ‘Changes of cancer cells to small cell cancer cannot be identified using current liquid biopsy technique’ – Does not make sense. What dose ‘small cell caner’ mean?
Response: As per your and reviewer 1 suggestion, we have removed the mentioned disadvantage and other disadvantages have been rewritten and appropriately cited.
4) Page 3, line 83: ‘Several studies have…” Need newer references for the statement (the current references were too old, from 1975, 1977).
Response: The text in line 83 have been removed in the modified manuscript.
5) Page 3, line 90-91: ‘A study conducted by Bettegowda C et al has demonstrated that ctDNA could be detected in patients without detectable CTCs’. Suggest changing to ‘A study conducted by Bettegowda C et al has showed that ctDNA could be detected in patients without images detectable CTCs’.
Response: As per the suggestion of reviewer 1, CTC as the source of cf/ctDNA, can be very puzzling for the reader, hence we have rephrased the clinical utility of CTC on the basis of CTC count as a molecular tumor target. Therefore, we have removed the literature work of Bettagowda C from the text.
6) Page 3, line 98-100: “There is currently… inform prognosis” Need a reference.
Response: A new reference was added to address the line (reference number 64).
7) Page 5, line 175 : ‘benign’ should be ‘non-metastatic’.
Response: The term “benign” is now replaced with “non metastatic” as per your suggestion.
8) Page 5, line 178-181: The sentence ‘A study on… disease[97]’ is confusion, needs to redo.
Response: As per your suggestions, lines were remodified as ‘A retrospective study done on 107 CTC positive metastatic breast cancer patients depicted that liquid biopsy could be useful method for revaluation of HER2 receptor status”.
9) Page 6, line 210-212: ‘Recurrence…As …Detection’ needs to redo
Response: As per your suggestions, lines were remodified as ‘Liquid biopsy can be used as a potential tool for detecting tumour recurrence as tumor-specific copy number aberrations persist to about 12 years after diagnosis’
Round 2
Reviewer 1 Report
The authors have greatly improved the manuscript. All open question were adressed and the manuscript was well restructured
Reviewer 2 Report
none.